# RETRIEVAL-AUGMENTED TEXT-TO-3D GENERATION

## ABSTRACT

Text-to-3D generation using using neural networks has been confronted with a fundamental difficulty regarding the scale and quality of 3D data. Score distillation sampling based on 2D diffusion models addresses this issue effectively; however, it also introduces 3D inconsistencies that plague generated 3D scenes due to a lack of robust 3D prior knowledge and awareness. In this study, we propose a novel framework for retrieval-augmented text-to-3D generation that is capable of generating superior-quality 3D objects with decent geometry. After we employ a particle-based variational inference framework, we augment the conventional target distribution in SDS-based techniques with an empirical distribution of retrieved 3D assets. Furthermore, based on the retrieved 3D assets, we propose the two effective methods: a lightweight adaptation of a 2D prior model for reducing its inherent bias toward certain camera viewpoints, and delta distillation to regularize artifacts of generated 3D contents. Our experimental results show that our method not only exhibits state-of-the-art quality in text-to-3D generation but also significantly enhances the geometry compared to the baseline.

## 1 INTRODUCTION

Text-to-3D generation becomes an crucial part of media production with the advances in deep generative models. However, compared with 2D images, the lack of high-quality and large-scale 3D data is an obstacle to train a text-to-3D generative model, while the scarcity of 3D data results from the intricate and labor-intensive process of 3D asset creation. Thus, the inherent difficulty of 3D data collection leads to limited quality and scale of 3D data despite successive releases of 3D datasets (Deitke et al., 2023a;b)

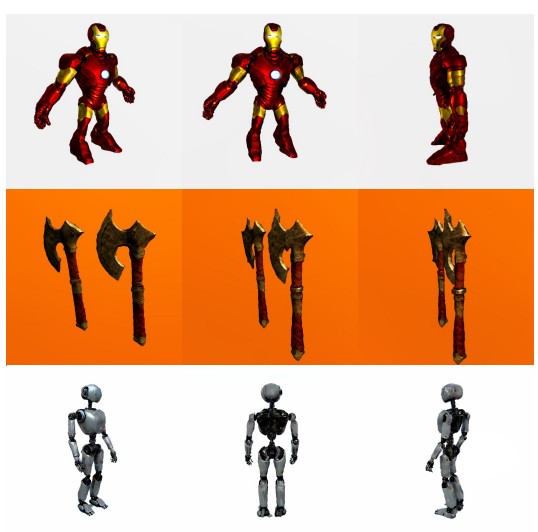

Previous approaches of text-to-3D generation leverage a text-to-image (T2I) model (Rombach et al., 2022a) trained on large-scale text-image pairs Schuhmann et al. (2022) to circumvent the scarcity of 3D data. For example, Zero123 (Liu et al., 2023b) modifies and fine-tunes a T2I model on multi-view datasets (Deitke et al., 2023a;b) to inject 3D-awareness into its parameters for the task of novel view synthesis. However, fine-tuning of large-scale 2D diffusion models requires expensive training costs, while the lack of high-quality 3D data lowers the fidelity of generated novel views.

Figure 1: Our framework generates high quality of 3D contents with geometric consistency by leveraging retrieved assets from external databases.

Meanwhile, Score Distillation Sampling (SDS) (Poole et al., 2022; Wang et al., 2023a) enables a 3D representation to be directly optimized via the prior knowledge of 2D diffusion models without any 3D data. SDS-based frameworks can leverage the capability of the T2I models to synthesize high-quality images to achieve high-fidelity of text-to-3D generation. Although recent studies (Chen et al., 2023; Wang et al., 2023b) significantly improve the high-fidelity results, a generated 3D con-

tent commonly suffers from artifacts and geometric inconsistency, since their optimization process relies solely on a 2D diffusion model, which lacks the awareness of 3D geometry.

To address these issues, we propose a framework for Retrieval-Augmented Text-to-3D generation to leverage the information of 3D data without full fine-tuning of 2D diffusion models. Our key insight is that semantically similar 3D assets to a given text can be a 3D geometric prior for SDS-based approaches, although existing 3D asset datasets (Deitke et al., 2023b) are inadequate for training a diffusion model in terms of scale and quality. Thus, we introduce an empirical distribution of retrieved 3D assets to augment the conventional SDS-based methods, which are originally based on the prior of 2D images only. Specifically, after we formulate a text-to-3D generation as a framework of variational score distillation (Wang et al., 2023b), we propose a novel formula to integrate the empirical distribution of retrieved 3D assets with the Particle-based Variational Inference (ParVI). Then, we also propose two novel technique to improve the geometric consistency and quality of generated 3D contents: i) we adopt a lightweight adaptation using the retrieved 3D assets to remove the bias of a 2D diffusion models on few camera viewpoints, ii) we propose a delta distillation to stabilize the optimization of 3D representations and reduce artifacts in generated 3D contents. The extensive empirical testing demonstrates the effectiveness of our retrieval-augmented text-to-3D generation on texture-fidelity and geometric consistency, compared with other baselines.

Our main contributions are summarized as follows: 1) Our retrieval-augmented text-to-3D generation effectively exploits both the geometric information of 3D assets and the capability of T2I models to synthesize high-fidelity images, instead of full training the model parameters. 2) We integrate the retrieval of 3D assets with the framework of ParVI for text-to-3D generation with VSD. 3) Based on the retrieved 3D assets, we propose Lightweight Adaptation to reduce the bias of T2I models on camera viewpoints and Delta Distillation to remove the artifacts. 4) The empirical results demonstrate that our proposed methods consistently improve the generation quality.

## 2 RELATED WORK

**Generative novel view synthesis.** Generative models can be used to learn a multi-view geometry to synthesize novel views of a 3D scene (Wiles et al., 2020; Rombach et al., 2021). Given a single reference view, Chan et al. (2023) estimate its 3D volume to condition a model for novel views. A cross-view attention is incorporated in a diffusion model to align the correspondences between novel views and the reference view (Zhou & Tulsiani, 2023; Watson et al., 2023). Zero123 (Liu et al., 2023b) modifies the Stable Diffusion model (Rombach et al., 2022a) to fine-tune its whole parameters on Objaverse (Deitke et al., 2023a;b) and generate novel views of 3D objects in the open domain. However, previous approaches have limited fidelity due to the scarcity of high-quality 3D data which requires a hand-craft work of experts.

**Text-to-3D generation with score distillation.** Poole et al. (2022) have proposed a novel method of Score Distillation Sampling (SDS) to generate 3D contents without 3D data, while optimizing a 3D representation such as NeRF (Mildenhall et al., 2021) via distilling the prior knowledge of diffusion models to synthesize high-fidelity images. Subsequent studies (Metzer et al., 2023; Tsalicoglou et al., 2023) have consistently improved text-to-3D generation based on SDS. Magic3D (Lin et al., 2023) exploits DMTet (Shen et al., 2021) with a coarse-to-fine pipeline to improve the quality of 3D representation. Fantasia3D (Chen et al., 2023) introduces a two-stage framework to disentangle geometry and texture of 3D contents. ProlificDreamer (Wang et al., 2023b) employs the particle-optimization framework for Variational Score Distillation (VSD) and significantly improves the fidelity of generated textures. However, these approaches without 3D training data commonly suffer from 3D inconsistency leading to unrealistic geometry of generated contents.

**Retrieval-augmented generative models.** Retrieval-augmented approaches utilize an external dataset to adapt a generative model for diverse tasks without fine-tuning whole parameters on large-scale data. For example, RETRO (Borgeaud et al., 2022) adapts a large language model for exploiting the external databases and achieves high performances without increasing its parameters. For the task of image synthesis, retrieval-augmented methods have been applied to GANs (Tseng et al., 2020; Casanova et al., 2021) and diffusion models (Blattmann et al., 2022; Sheynin et al., 2022; Chen et al., 2022), while adapting the models for synthesizing unseen styles such as artistic images Rombach et al. (2022c). Since retrieval-augmentation is effective when the data scale is insufficient to

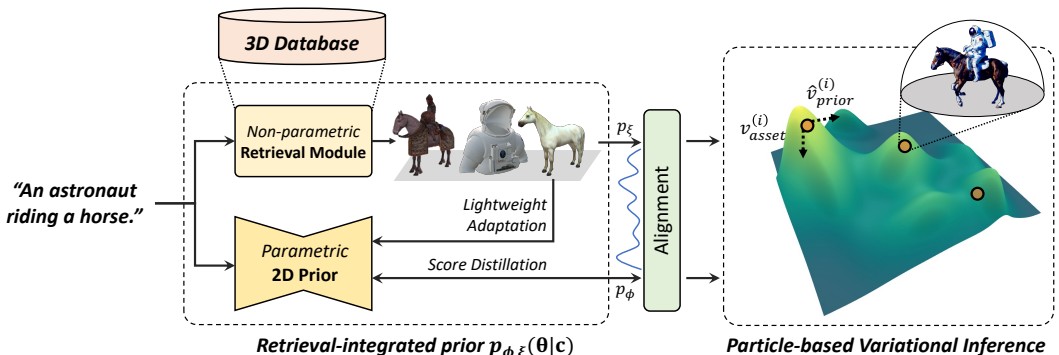

Figure 2: **Overview.** Given a prompt $c$, we retrieve the nearest neighbor from the 3D database. With these assets, we perform lightweight adaptation for better 2D prior, and we optimize a variational distribution by leveraging information from the adapted 2D prior and the retrieved 3D assets both.

train the model parameters, Zhang et al. (2023) and He et al. (2023) integrate a motion-retrieval module with diffusion models to synthesize motion sequences and videos, respectively.

## 3  PRELIMINARIES: TEXT-TO-3D GENERATION WITH 2D DIFFUSION PRIOR

In this section, we formulate text-to-3D generation with score distillation (Poole et al., 2022), which leverages a text-to-image diffusion model (Saharia et al., 2022; Rombach et al., 2022b) as the prior to optimize a 3D representation for a given text. We employ the framework of Variational Score Distillation (VSD), since VSD is a general framework for text-to-3D generation Wang et al. (2023b). In fact, VSD aims to optimize the distribution of 3D representations given a text prompt, while SDS aims to optimize an instance of 3D representation for text-to-3D generation (see Appendix). We denote $\gamma(\theta|c)$ as a distribution of 3D representations $\theta$ on a text prompt $c$, and $g(\theta, \psi)$ as a rendering function for a 3D representation $\theta$ and a camera viewpoint $\psi$. We also define $q^\gamma(x|c, \psi)$ as an implicit distribution of the rendered image $x := g(\theta, \psi)$ where $\theta \sim \gamma(\theta|c)$. Then, VSD for text-to-3D generation minimizes the variational objective, $D_{\mathrm{KL}}\big(q^\gamma(x|c)||p_\phi(x|c)\big)$ to find an optimal $\gamma^*$, where $q^\gamma(x|c)$ is marginalized distribution w.r.t. camera viewpoints $p(\psi)$ and $p_\phi(x|c)$ is empirical likelihood of $x$ estimated by a diffusion moddel $\phi$. Since the diffusion model learns noisy distribution $p_\phi(x_t|c, t)$ according to diffusion process (Ho et al., 2020; Song et al., 2020), the variational objective can be decomposed as follows:

$$\gamma^* := \arg\min_\gamma \mathbb{E}_t\Big[(\sigma_t/\alpha_t)w(t)D_{\mathrm{KL}}(q_t^\gamma(x_t|c)||p_\phi(x_t|c, t))\Big], \tag{1}$$

where $q_t^\gamma(x_t|c)$ is a noisy distribution at noise level $t$ following the diffusion process.

VSD exploits the particle-based variational inference (ParVI) (Chen et al., 2018; Liua & Zhub, 2022; Wang et al., 2019; Dong et al., 2022) to minimize Eq. 1. The minimization process proceeds via a Wasserstein gradient flow (Chen et al., 2018) in the ParVI framework. Specifically, $N$ particles $\{\theta^{(i)}\}_{i=1}^N$ are first sampled from initial $\gamma(\theta|c)$, and then updated with the following ODE:

$$v_{\mathrm{prior}} := \frac{d\theta_\tau}{d\tau} = -\mathbb{E}_{t,\epsilon,\psi}\Big[w(t)\big(-\sigma_t\nabla_{x_t}\log p_\phi(x_t|c, t) - (-\sigma_t\nabla_{x_t}\log q_t^{\gamma_\tau}(x_t|\psi, c))\frac{\partial g(\theta_\tau, \psi)}{\partial\theta_\tau}\big)\Big], \tag{2}$$

where $\tau$ denotes ODE time such that $\tau \geq 0$, and the distribution $\gamma_\tau$ converges to an optimal distribution $\gamma^*$ as $\tau \to \infty$ and $\theta_\tau$ is sampled from $\gamma_\tau$. Note that the first term is a score of noisy real image, approximated by a predicted score of the diffusion model $\epsilon_\phi(x_t, c, t)$. The second term can be regarded as a score of noisy rendered images. They parameterize the second term to a score-predicting U-shaped network. Practically, they train the U-Net network from the pretrained diffusion model with low-rank adaptation (LoRA) (Ryu, 2023): $-\nabla_{x_t}q_t^{\gamma_\tau}(x_t|\psi, c) = \epsilon_{(\phi,\zeta)}(x_t, t, c, \psi)$, where $\zeta$ is a set of parameters of trainable residual layers for LoRA. We attach the details of derivations regarding VSD and ParVI to Appendix. VSD allows to generate realistic textures of 3D object given a text, but we remark that these method are still vulnerable to generating unrealistic geometry.

# 4 RETRIEVAL-AUGMENTED 3D GENERATION

While previous methods based on SDS allow for the flexible generation of 3D objects, even with complex user prompts, they tend to produce implausible 3D geometry. In this work, we propose a novel approach requiring a minimal amount of training, capable of generating 3D objects of superior quality. Specifically, we adopt a retrieval-based approach that ensures the creation of high fidelity, view-consistent 3D contents. This effectively mitigates the drawbacks encountered in previous strategies that rely solely on 2D prior models or necessitate large-scale training on 3D assets.

To realize this goal, we explore methodologies for both the integration of knowledge from retrieved 3D assets and the process of 3D retrieval itself. In Section 4.1, a new variational objective is proposed, which incorporates knowledge from these retrieved assets as well as from 2D prior models. Following this, in Section 4.2, we present a method that adapts 2D prior models using the retrieved 3D assets, with the aim of reducing bias toward certain camera viewpoints inherent in the 2D prior. Furthermore, in Section 4.3, we propose a simple yet effective delta distillation technique to regularize artifacts. Finally, in Section 4.4, we illustrate the retrieval of assets from a 3D dataset which utilizes both text captions and rendered images. The overview of our whole pipeline is in Fig. 2

## 4.1 RETRIEVAL-INTEGRATED OBJECTIVE

Let $\xi_N(c, \mathcal{D})$ be a non-parametric sampling strategy to obtain the $N$ nearest neighbors using the retrieval algorithm conditioned on text prompt $c$ in the 3D dataset $\mathcal{D}$. Our goal is to integrate the rich view-dependent information from the retrieved assets with that of 2D prior models, and derive the particle-based optimization process for the variational distribution $\gamma(\theta|c)$.

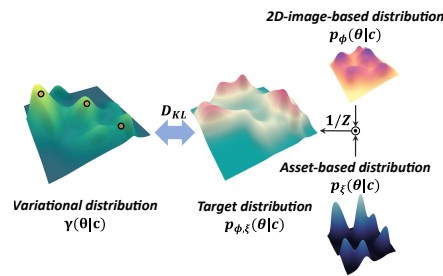

Figure 3: **Conceptual figure of the variational objective.** Geometrically plausible areas by retrieved nearest neighbors have higher density in the target distribution.

We assume the probability density of 3D content $\theta$ by 2D prior is proportional to the expected densities of its multiview images w.r.t. camera viewpoints, following Wang et al. (2023a):

$$p_\phi(\theta|c) \propto \mathbb{E}_\psi\big[p_\phi^{\text{2D}}(x|c, x = g(\theta, \psi)\big]. \quad (3)$$

Technically, this expectation is set as the geometric expectation (see Appendix D for details).

In this section, we introduce a novel energy functional for integrating the retrieved assets:

$$\mathcal{E}[\gamma] := D_{\text{KL}}\big(\gamma(\theta|c)||p_{\phi,\xi}(\theta|c)\big), \quad (4)$$

where we present $p_{\phi,\xi}(\theta|c)$ as a retrieval-integrated prior. Based on the intuition that a 3D asset selectively filters a distribution, we simply multiply and normalize the two distributions:

$$p_{\phi,\xi}(\theta|c) := \frac{1}{Z'} p_\phi(\theta|c) p_\xi(\theta|c), \quad (5)$$

where we denote $p_\xi(\theta|c)$ as a 3D likelihood from the retrieved assets, and $Z'$ denotes the normalizing constant. Fig. 3 depicts the intuition behind this; the distribution $p_\xi$ derived from the retrieved nearest neighbor serves as an implicit filter for plausible geometry.

Specifically, we derive the distribution $p_\xi(\theta|c)$ from an empirical distribution defined over the top-$N$ nearest neighbors $\{\theta_{\text{ret}}^{(n)}\}_{n=1}^N$ utilizing the sampling strategy $\xi_N(c, \mathcal{D})$, then applying Gaussian kernel $K_\sigma$ with a variance $\sigma^2$. Intuitively, the likelihood $p_\xi(\theta|c)$ depicts how close the particle is to the retrieved assets.

Using the definition of KL divergence (as detailed in Appendix D), this is further expanded:

$$\mathcal{E}[\gamma] = \mathbb{E}_\psi[D_{\text{KL}}\big(q^\gamma(x|c)||p_\phi^{\text{2D}}(x|c)\big)] + H\big(\gamma(\theta|c); p_\xi(\theta|c)\big) - C \quad (6)$$

$$= \mathbb{E}_\psi[D_{\text{KL}}\big(q^\gamma(x|c)||p_\phi^{\text{2D}}(x|c)\big)] - \mathbb{E}_{\gamma(\theta|c)}\left[\log \sum_n K_\sigma(\theta - \theta_{\text{ret}}^{(n)})\right] - C', \quad (7)$$

where $x = g(\theta, \psi)$, and $H$ is the joint entropy. $C$ and $C'$ are constants to be unnecessary.

The minimization process then proceeds via a Wasserstein gradient flow (Chen et al., 2018). Given $\mathcal{E}[\gamma_\tau]$ at an optimization step $\tau$, the velocity of particles, $v_\tau := \frac{d\theta_\tau}{d\tau} = \nabla_\theta \frac{\delta\mathcal{E}[\gamma_\tau]}{\delta\gamma_\tau}$, is obtained by calculating the functional derivative $\frac{\delta\mathcal{E}[\gamma_\tau]}{\delta\gamma_\tau}$ as follows:

$$v_\tau = \nabla_\theta \frac{\delta\mathcal{E}[\gamma_\tau]}{\delta\gamma_\tau} = v_{\text{prior}} - \nabla_\theta \log \sum_n K_\sigma(\theta - \theta_{\text{ret}}^{(n)}) \tag{8}$$

$$= v_{\text{prior}} + v_{\text{asset}}. \tag{9}$$

where $v_{\text{asset}}$ is the velocity derived from retrieval, and $v_{\text{prior}}$ is derived as in Eq. 2.

However, directly computing the derived $v_{\text{prior}}$ remains inefficient, given that it is defined in a high-dimensional space. To address this inefficiency, we turn to our empirical observations, which suggest a feasible alternative. We observe that the direction of the velocity of each particle is largely determined by its random initialization, as it is drawn towards the nearest mode (see Appendix E for details). Motivated by this observation, instead of computing all terms, we use an efficient surrogate method to compute $v_{\text{asset}}$ for each particle as follows:

$$v_{\text{asset}}^{(i)} = \sum_n \frac{\pi_n^{(i)}}{\sigma^2}(\theta^{(i)} - \theta_{\text{ret}}^{(n)}) = \frac{1}{\sigma^2}\sum_n \pi_n^{(i)}(\theta^{(i)} - \theta_{\text{ret}}^{(n)}), \tag{10}$$

where $\theta^{(i)}$ is $i$-th particle from the variational distribution $\gamma(\theta|c)$ and we assign to them one-hot vectors $\pi$ whose non-zero indices correspond to a closest random asset when initialized. Intuitively, this property of a particle to follow a specific mode is determined at the time of its creation.

For generality, the particle $\theta^{(i)}$ and 3D asset $\theta_{\text{ret}}^{(n)}$ have not been assumed to have specific representations (*e.g.*, NeRF (Mildenhall et al., 2021), DMTet (Shen et al., 2021), or mesh), and could be different representations. However, some representations can be only partially observed through the differentiable rendering function $g$. Accordingly, in Eq. 10, the shift term is given in the form of a gradient with respect to the objective (Tancik et al., 2021):

$$(\theta^{(i)} - \theta_{\text{ret}}^{(n)}) \simeq \nabla_{\theta^{(i)}} \mathbb{E}_\psi\left[\left\|g(\theta^{(i)}, \psi) - g(\theta_{\text{ret}}^{(n)}, \psi)\right\|_2^2\right]. \tag{11}$$

## 4.2 LIGHTWEIGHT ADAPTATION OF 2D PRIOR MODELS WITH RETRIEVED 3D ASSETS

Even though our process integrates 3D information from assets, we find it beneficial to reduce the bias toward certain camera viewpoints of the 2D prior $p_\phi^{2\text{D}}(x|c)$ as shown in Fig. 4, since these are identified as a factor contributing to the view inconsistency problem. Specifically, to achieve this goal, we introduce a lightweight strategy that adapts 2D prior models by utilizing retrieved 3D assets. This helps balance the probability densities across all viewpoints without a significant drop in the quality of the original 2D prior models.

We denote $c_{\text{ret}}^{(i)}$ as a ground-truth text caption corresponding to the $i$-th retrieved asset $\theta_{\text{ret}}^{(i)}$, and $e(c, \psi)$ as a function that outputs a text caption $c$ with a view augmentation (Poole et al., 2022) that matches the camera viewpoint $\psi$. To obtain a adapted 2D prior $\epsilon_{(\phi,\omega)}$, we densely render the retrieved assets under a uniform camera distribution and perform a low-rank adaptation (LoRA) (Ryu, 2023) with the rendered images:

$$\min_\omega \sum_{i=1}^N \mathbb{E}_{t,\epsilon,\psi}\left\|\epsilon_{(\phi,\omega)}\left(\alpha_t g(\theta_{\text{ret}}^{(i)}, \psi) + \sigma_t\epsilon, t, e(c_{\text{ret}}^{(i)}, \psi)\right) - \epsilon\right\|_2^2, \tag{12}$$

where $\omega$ is a set of parameters of learnable layers inserted to the diffusion U-net for low-rank adaption. Note that $\phi$ is a set of parameters of the diffusion U-Net. As shown in Fig. 4, samples from the adapted 2D prior reflect a more diverse range of viewpoints that correspond to the view condition, without severely sacrificing the original generation capability.

## 4.3 DELTA DISTILLATION FOR REGULARIZING ARTIFACTS

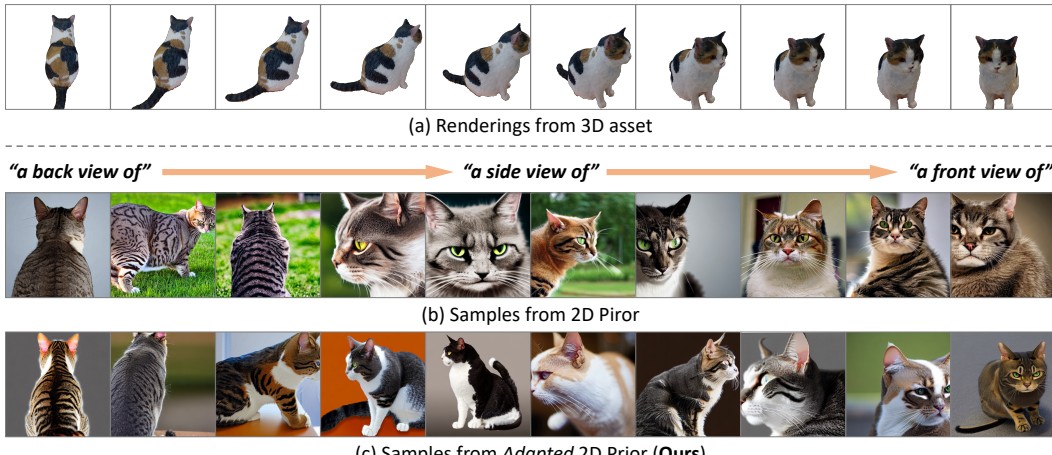

(a) Renderings from 3D asset

*"a back view of"* ⟶ *"a side view of"* ⟶ *"a front view of"*

(b) Samples from 2D Piror

(c) Samples from *Adapted* 2D Prior (**Ours**)

Figure 4: **Lightweight adaptation of 2D diffusion models.** We compare the effectiveness of the adaptation with given rendering from a 3D asset in (a). We linearly interpolate a text embedding from "*a back view of an angry cat*" to "*a front view of an angry cat*" through "*side view*". Compared with (b). The samples from adapted 2D prior in (c) reflect a variety of viewpoints, not biased towards a single viewpoint. We provide more results in Appendix. B.1.

The co-formulation proposed in the preceding section facilitates a retrieval-augmented approach for text-to-3D generation. This can be interpreted as sampling from the combined distribution of the 2D diffusion prior and the retrieved assets, each with their respective velocities. In practice, however, we notice that the conflicting effects of two independent velocities from different sources often generate artifacts, as illustrated in Fig. 5.

To address this issue, we propose a delta distillation technique, which subtracts the component deemed to be unreliable, inspired by Hertz et al. (2023). Specifically, we denote $v_{\mathrm{prior}}(\theta = \theta_0)$ as the predicted velocity (gradient) of the 2D prior at the point $\theta_0$. Since a pair of a retrieved asset and the text should ex-

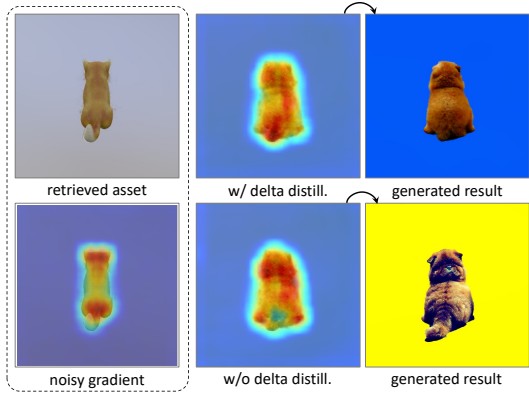

Figure 5: **Motivation of delta distillation.** We subtract a noisy gradient to regularize artifacts.

hibit minimal gradient or velocity if the retrieval is successful, we characterize $v_{\mathrm{prior}}(\theta = \theta_{\mathrm{ret}})$ as an unconditional, noisy velocity. To mitigate the artifacts, we subtract this from the original $v_{\mathrm{prior}}$:

$$\hat{v}_{\mathrm{prior}} \coloneqq v_{\mathrm{prior}} - v_{\mathrm{prior}}(\theta = \theta_{\mathrm{ret}}). \tag{13}$$

Subsequently, we employ $\hat{v}_{\mathrm{prior}}$ instead of $v_{\mathrm{prior}}$ in Eq. 9. Empirically, we observed that this omission is pivotal for stable generation performance and for the reduction of artifacts.

## 4.4 3D DATASET RETRIEVAL

We collect 3D assets from Objaverse 1.0 (Deitke et al., 2023b) dataset which contains diverse set of over 800K+ 3D assets. We retrieve $N$ nearest neighbors using ScaNN (Guo et al., 2020) vector search algorithm using CLIP embeddings (Radford et al., 2021). The query embedding can be acquired from either $c$, which is the prompt for the diffusion model, or from the prompt additionally given by the user. The CLIP features for the database can be obtained with the corresponding captions or rendered images. For the captions of the 3D assets, we utilize Cap3D (Luo et al., 2023) dataset which contains predicted captions of 3D objects in Objaverse.

Either captions and rendered images can be utilized. We observed distinct characteristics for each. For the embedding space of text captions, objects in a category related to the query embedding are

effectively retrieved with robust performance. In contrast, for the embedding space for rendered images, the texture information of the query embedding is taken into sufficient consideration, but it tends to have relatively frequent failure cases. In Fig. 6, we show the retrieved data using caption and rendered image. Hence, we utilize both image and text embeddings by performing Top-K operation with image embeddings after retrieving $N'(N' > N)$ objects with text embeddings.

Although Objaverse dataset provides diverse 3D assets, their orientations are not aligned. Therefore, before we put our nearest neighbors to use, we must align their frontal view. To this end, we calculate the CLIP similarity score between the prompts "*front view*", "*side view*", "*back view*" and the rendered images with different camera poses. Then, we rotate the 3D assets with the camera pose having the relatively highest CLIP similarity score. Despite its simplicity, this method is capable of aligning our retrieved assets effectively.

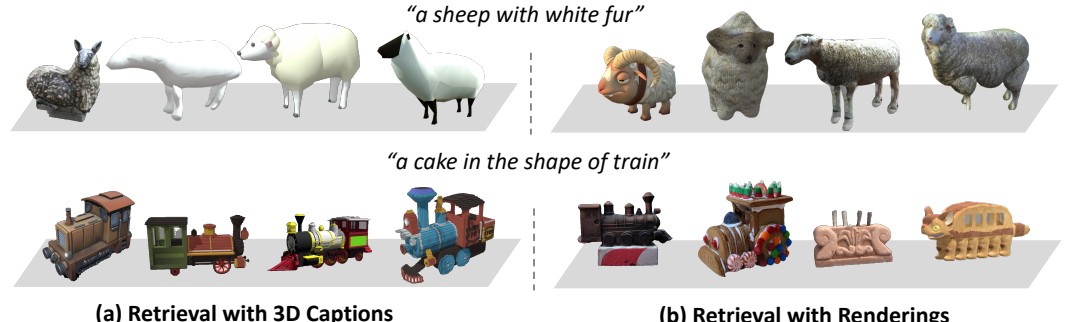

*"a sheep with white fur"*

*"a cake in the shape of train"*

**(a) Retrieval with 3D Captions**    **(b) Retrieval with Renderings**

Figure 6: **3D Dataset retrieval.** (a) and (b) show retrieved top-$K$ nearest neighbors on CLIP-text embedding space and CLIP-image embedding space, respectively.

## 5 EXPERIMENTS AND DISCUSSION

### 5.1 IMPLEMENTATION DETAILS

We build our method on the implementation of ProlificDreamer (Wang et al., 2023b) from Three-studio (Guo et al., 2023). Instant-NGP (Müller et al., 2022) is used for our NeRF backbone and Stable Diffusion v2 (Rombach et al., 2022b) as the 2D prior. For all experiments, we retrieve 3 assets in Objaverse 1.0 (Deitke et al., 2023b). We render our retrieved data with 100 uniformly sampled camera poses, using Blender. all hyperparameters including the number of particles follow our baseline, the implementation of ProlificDreamer (Wang et al., 2023b; Guo et al., 2023), except newly introduced hyperparameters such as $\sigma$.

### 5.2 RETRIEVAL-AUGMENTED 3D GENERATION

**Qualitative evaluation.** We show qualitative result of our method in Fig. 1. We then compare our method with our baseline, ProlificDreamer (Wang et al., 2023b). In Fig. 7, note that ProlificDreamer shows unwanted artifacts as well as geometric inconsistencies *e.g.*, multiple heads. However, ours generates cleaner and consistent 3D model while maintaining the high textural quality of Prolific-Dreamer. We additionally compare our method with novel view generative model, Zero123 (Liu et al., 2023a). As Zero123 takes an image, not text prompt, we first generate an image for conditioning with Stable Diffusion model and preprocess it. Then, we apply SDS for 3D generation. As shown in Fig. 8. Zero123 tends to generate simplified texture and the performance degrades when given a realistic image as shown in the two rightmost columns. More results with other methods including Dreamfusion Poole et al. (2022), and Magic3D Lin et al. (2023) are in the Appendix A.

**3D consistency.** To demonstrate the 3D consistency of our approach, we evaluate the adherence to the view prompt, "*front view*" and "*back view*", in Fig. 9. Specifically, we calculate the relative similarity, using the CLIP score between the rendered image and view prompts corresponding to three perspectives: "*front view*" "*side view*" and "*back view*". For example, for the relative similarity of "*front view*", we divide its CLIP similarity score with the sum of similarity scores of the rendered image and the other view prompts. Assuming high 3D consistency yields high relative CLIP simi-

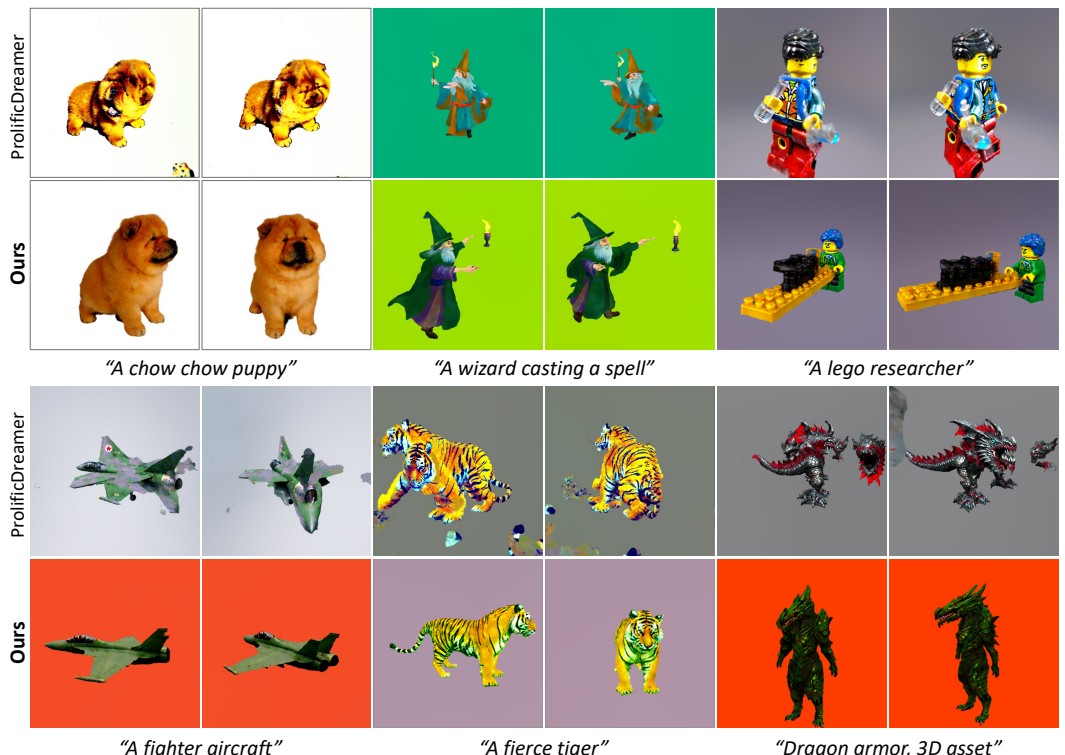

Figure 7: **Qualitative comparison with ProlificDreamer (Wang et al., 2023b).** Our method achieves higher geometric consistency compared to Prolificdreamer. Comparison with other methods are in the Appendix A

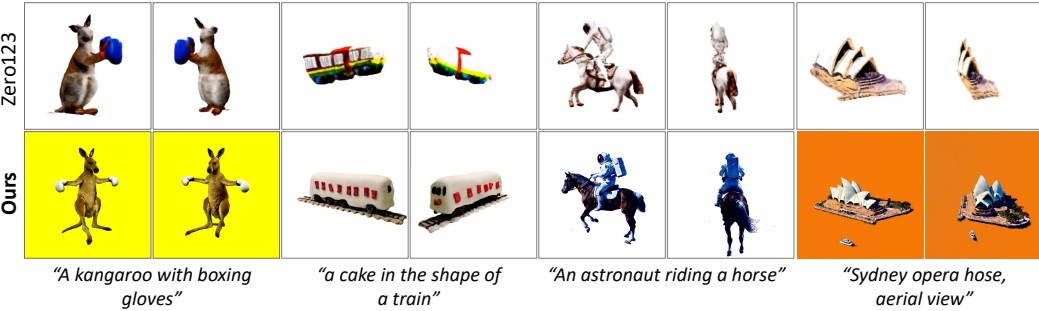

Figure 8: **Comparison with Zero123 (Liu et al., 2023a).** We compare our model with Zero123, a state-of-the-art novel view generative model, conditioned on a single image. Our method generates 3D content with better fine details and geometry.

larity score between the rendered image and the view prompt associated with its specific azimuth, we show the relative CLIP similarity scores for images rendered from our method and the baseline, each rendered at various azimuths, same elevation.

**User study.** We conducted a user study with 92 participants; the result is shown in Fig. 10. Each participant asked seven randomly selected questions. Specifically, we inquired about their preference between our method and the baseline, taking into account geometry and textural fidelity. Approximately 75% of the participants expressed a preference for the results by our method over the baseline. More details are described in Appendix C.

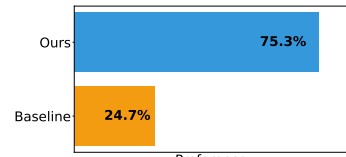

Figure 10: **User preference study.**

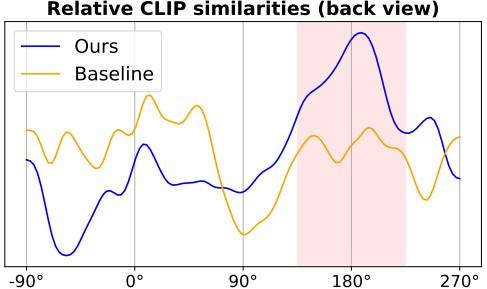

Figure 9: **Relative CLIP Similarities of rendered images.** (Left) Relative similarity between ours and baseline (Wang et al., 2023b) with "front view". (Right) Relative similarity with "back view". The red shaded areas represent ±45 degrees from the front and back respectively. Our method exhibits higher relative similarity in the region that matches the view prompt.

**Impact of Retrieved 3D Assets** To inspect the impact of the retrieved assets and the generated results, we show the first nearest neighbor from the retrieved assets with the result of the corresponding particle in Fig. 11. We observe that particle converges at the point where it aligns with both the prompt and the retrieved data. Specifically, in Fig. 11, the position of the ball and the dog in the resulting particle of the first row aligns with its nearest 3D pair while the legs are deformed for textual alignment with the prompt.

**Ablations.** In Fig. 4 and Appendix B.1, we provide the ablation on manifold tweaking. Adapting the 2D prior model with densely rendered image of 3D asset mitigates the internal bias towards frontal view of the 2D prior, showing various viewpoints. Additionally, we conduct ablations on delta score distillation and multiple particle setting in Appendix B.2 and Appendix B.3 respectively. Specifically, we show that Delta Distillation reduces floating artifacts and improves geometric awareness. Also, we increase the number of particles and present the following result.

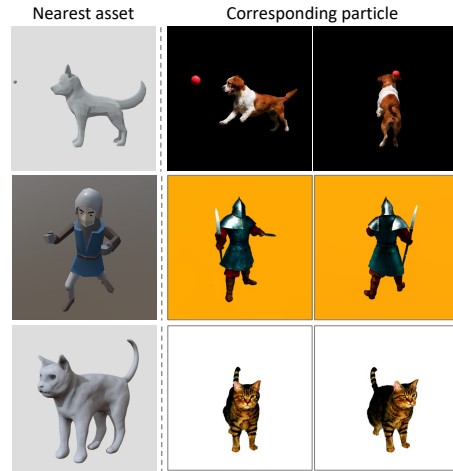

Figure 11: **Nearest neighbor and the result of corresponding particle.** The given prompts, starting from the top, are "*a dog playing fetch*", "*a medieval knight*", and "*an angry cat*".

## 6 CONCLUSION

We introduce a novel methodology that requires minimal training, yet is proficient in creating state-of-the-art-quality 3D objects. By embracing a retrieval-centric approach, we ensure the creation of high-fidelity and view-consistent 3D content, effectively overcoming the limitations seen in previous frameworks that were either heavily reliant on 2D prior models or required extensive training on 3D assets. A new variational objective is introduced, facilitating a smooth integration of knowledge from both the retrieved assets and 2D prior models. Following this, we propose a lightweight adaptation of 2D prior models as well as novel delta distillation technique to balance the density across viewpoints and reduce artifacts. Lastly, the mechanisms of retrieving assets from a 3D dataset are explained through the utilization of text captions alongside rendered images. With comprehensive experiments and discussion, we demonstrate the effectiveness of our methods in improving the view-consistency as well as retaining the superior quality, and pave the road for text-to-3D methods that capture both the fidelity and geometry. The potential applications of this methodology span various domains such as virtual reality, video game design, film production, and architectural visualization, where the rapid and high-quality generation of 3D content is crucial.

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

# A    ADDITIONAL QUALITATIVE COMPARISIONS WITH OTHER METHODS

Figure 12: **Qualitative comparisons with other methods (Poole et al., 2022; Lin et al., 2023).** The additional comparison with Dreamfusion (Poole et al., 2022), and Magic3D (Lin et al., 2023) can be found in Fig. 12. We use the implementation from threestudio Guo et al. (2023) as the official code has not been released. Our method shows better texture and geometry compared to other two methods.

# B    ABLATION STUDY

## B.1    ABLATION ON LIGHTWEIGHT ADAPTATION

Fig. 13 shows additional result on lightweight adaptation. The images in the top row are used during adaptation process. The generated results from plain 2D prior show biased result to the frontal view of the object while the adaptation method shows high fidelity to the view prompt.

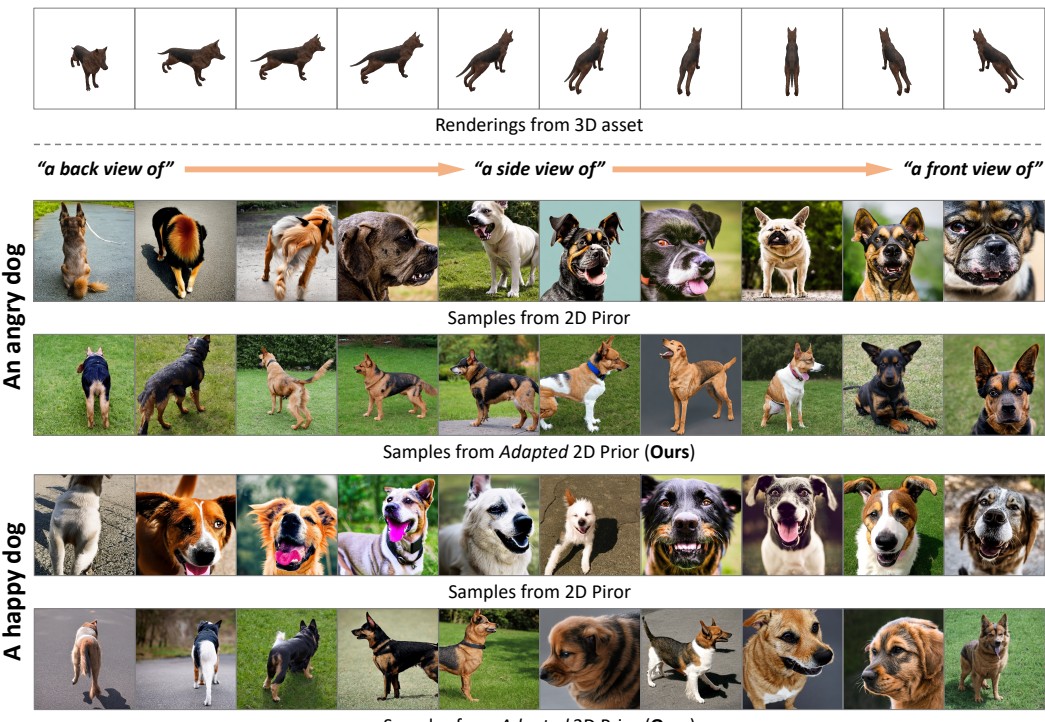

Figure 13: **More results of ablation on lightwight adaptation.**

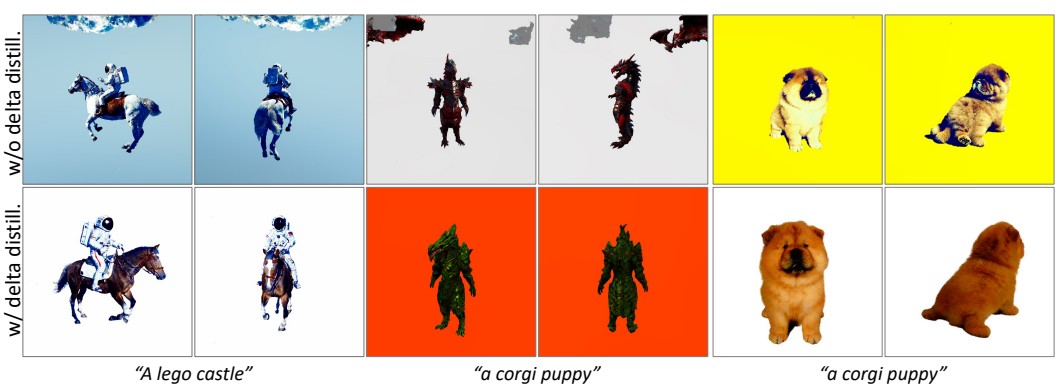

Figure 14: **Results of ablation on delta distillation.**

## B.2 ABLATION ON DELTA DISTILLATION

We show the effectiveness of the Delta Distillation in Fig. 14. The result generated with Delta Distillation shows unwanted artifacts (left two prompts) and multi faces (right prompt) while ours which utilizes Delta Distillation shows superior results.

## B.3 ABLATION ON NUMBER OF MULTI-PARTICLES

We increase the number of particles, each assigned to one of the retrieved asset. The result generated with three particles is shown in Fig. 15. The figure shows diverse result generated with the same prompt.

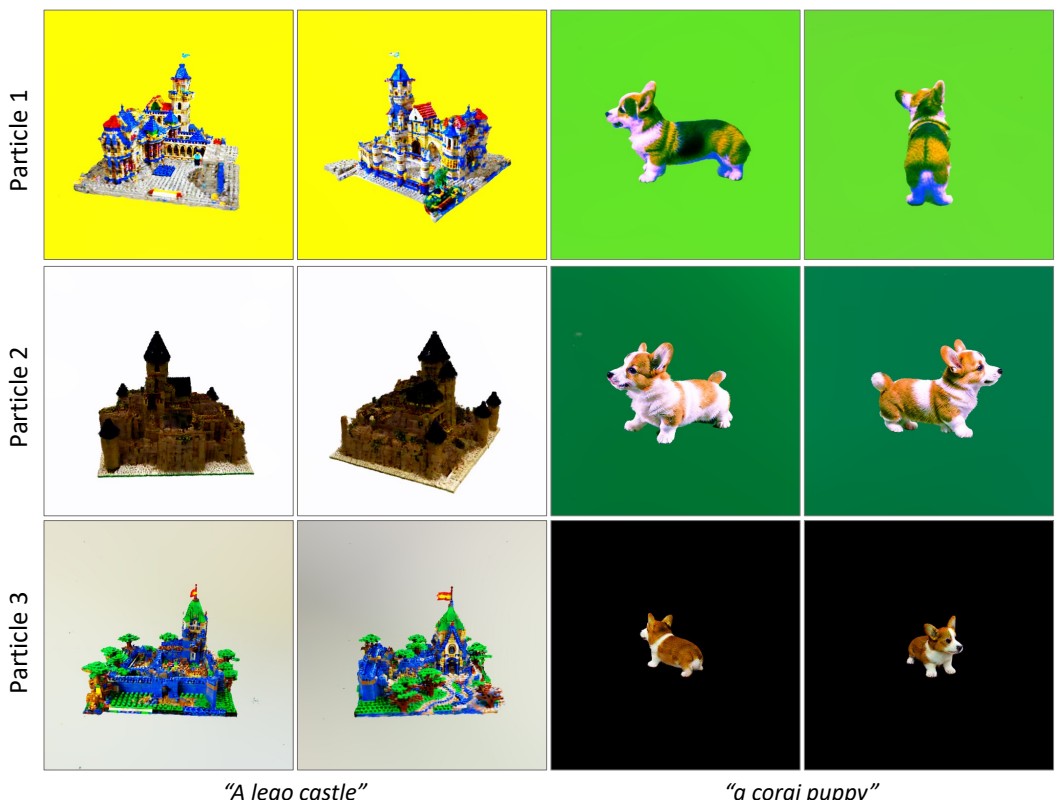

Figure 15: **Generated results with thee particles.**

## C  USER STUDY

We conduct user to qualitively compare our method with the baseline Wang et al. (2023b). Given a 360° video of the generated scene, the participants are asked to answer the following question, 'When considering both texture and shape (geometry), which result is more satisfying?'. Each participant was asked about a randomly selected set of six generated results and prompt pairs, with their anonymity being ensured. The reported statistics summarizes responses from a total of 92 participants.

## D  ASSUMPTION ON THE DENSITY FUNCTION OF 3D CONTENT

Several works Wang et al. (2023a); Hong et al. (2023) have clarified the assumptions on the density function of 3D content, which is an important part in lifting the 2D generative models to do 3D generation. Specifically, SJC Wang et al. (2023a) proposes to assume it to be proportional to an arithmetic expectation of likelihoods over camera points, *i.e.*, $p_\phi(\theta|c) \propto \mathbb{E}_\psi[p_\phi^{2D}(x|c, x = g(\theta, \psi))]$, and D-SDS Hong et al. (2023) finds it more beneficial to define it as a product of likelihoods over a set of camera points.

In this paper, we instead use the geometric expectation. Actually, all three premises do not affect the solution of the minimization or maximization problem of the logarithm. Besides, in terms of KL divergence, setting the target distribution to the geometric mean has the following benign property:

$$D_{KL}(q||\kappa \mathbb{G}_\psi[p_\phi^{2D}(x|c, x = g(\theta, \psi))) = \mathbb{E}_\psi[D_{KL}(q||p_\phi^{2D}(x|c, x = g(\theta, \psi)))] - \log \kappa, \quad (14)$$

where $\kappa$ is a constant.

# E  ELABORATION ON THE OBSERVATION IN SEC. 4.1

When $\sigma$ is small enough, the behavior of the Gaussian kernel $K_\sigma$ becomes peaked. Specifically, $K_\sigma(\theta - \theta_{\text{ret}}^{(k)})$ gets very high at values of $k$ where $\theta$ is nearly equal to $\theta_{\text{ret}}^{(k)}$, and decreases rapidly as $\theta$ moves away from $\theta_{\text{ret}}^{(k)}$. As a result, the sums in the expression are mostly influenced by the terms where $k$ value makes $\theta_{\text{ret}}^{(k)}$ closest to $\theta$, denoted as $\theta_{\text{ret}}^{(k_{\min})}$.

Mathematically, the exponential term in the Gaussian kernel heavily outweighs the term $(\theta - \theta_{\text{ret}}^{(k)}/\sigma^2)$ in the formula. The latter term changes more slowly with respect to $\theta - \theta_{\text{ret}}^{(k)}$ compared to the exponential decay caused by the Gaussian kernel.

Therefore, the expression for $v_{\text{asset}}$ becomes mostly affected by the term corresponding to $k_{\min}$. This leads to the simplification $v_{\text{asset}} \approx \omega \frac{\theta - \theta_{\text{ret}}^{(k_{\min})}}{\sigma^2}$, where $\omega$ is a weighting factor. The term $\theta_{\text{ret}}^{(k_{\min})}$ denotes the point closest to $\theta$, which becomes the main contributor to the value of $v_{\text{asset}}$.

