# OpenReview forum: "Retrieval-augmented Text-to-3D Generation"
_ICLR.cc/2024/Conference — ICLR 2024 Conference Withdrawn Submission_

### Official Review · Reviewer_aEzM · 2023-10-31

**Soundness:** 2 fair
**Presentation:** 2 fair
**Contribution:** 2 fair
**Rating:** 5
**Confidence:** 5

**Summary:**

This paper presents a Retrieval-Augmented Text-to-3D generation framework that uses an empirical distribution of retrieved 3D assets. The method incorporates geometric information from retrieved 3D assets based on the text prompt from existing 3D asset datasets to improve the geometric quality and consistency of the generated 3D shapes. The study introduces two techniques: Lightweight Adaptation, addressing biases in camera viewpoints, and Delta Distillation, targeting 3D content artifacts.

**Strengths:**

1. The study proposes an approach to Text-to-3D generation that leverages retrieved 3D assets, potentially improving the quality and geometric consistency of generated objects.
2. The introduction of Lightweight Adaptation and Delta Distillation techniques aims to reduce biases from camera viewpoints and diminish artifacts in the 3D content, which have been challenging in previous models.

**Weaknesses:**

1. The experiments provided seem to be limited and do not fully substantiate the claims made. The supplementary materials only provided the videos of the generated objects with the same text prompts as in the paper.
2. The generation diversity could be an issue, for example in Figure 15, the three particles of the generated corgi are very similar.
3. The idea of Lightweight Adaptation is not new and the idea of Delta Distillation is mainly adopted from the Delta denoising score[1].
4. The presentation of the methodology part is not clear and intuitive, it seems the author tried to overcomplicate the narratives.
5. The authors have chosen not to disclose specific details about how they transform the retrieved 3D assets into the 3D shape representations \theta to acquire the Asset-based distribution. This process could be time and computationally intensive.



[1] Hertz, Amir, Kfir Aberman, and Daniel Cohen-Or. "Delta denoising score." Proceedings of the IEEE/CVF International Conference on Computer Vision. 2023.

**Questions:**

1. In section 4.1, how do you transform the retrieved mesh into the 3D representations?
2. What about the efficiency of this method? It seems to require a lot of time and computational resources to generate a single object from a text prompt.

---

### Official Review · Reviewer_U742 · 2023-10-31

**Soundness:** 2 fair
**Presentation:** 2 fair
**Contribution:** 3 good
**Rating:** 5
**Confidence:** 4

**Summary:**

This paper addresses the problem of Text-to-3D generation building on top of the ProlificDreamer variational score distillation (VSD) by integrating retrieved 3D assets into the model optimization procedure. In particular the VSD formulation try to match a parametric distribution of 3D shapes conditioned on a prompt to the distribution of 2D views generated by a text-to-image prior, the authors propose to integrate in the formulation another prior that relies on existing 3D assets retrieved from Objaverse to push the learned distribution of 3D shapes to be more 3D consistent and have less spurious artifacts (e.g., multiple frontal views). Besides being used explicitly in the optimization, rendering from 3D assets are also used to fine tune using LoRA the text-to-image prior to make it more aware of camera views description in the prompt. The result is a text-to-3D model that can generate more faithful 3D models that don’t show multiple repetitions of frontal views. Evaluation is mostly limited to qualitative examples and user studies.

**Strengths:**

+ The integration of the 3D assets prior into the VSD formulation is quite elegant and clearly helps to generate 3D assets with a more plausible 3D structure and artifacts free. The author provides a detailed mathematical justification for their method, explaining assumptions they consider while developing the final proposed method.

+ The fine tuning of the 2D diffusion prior using 3D assets is a sound idea that clearly seems to help to build 3D awareness into the model. Many previous work did not consider improving this specific aspect of the pipeline and I found interesting the focus of the authors on this aspect.

**Weaknesses:**

## Major

a.**Experimental evaluation**: In my opinion the experimental evaluation of the proposed method is very weak for two main aspects:
1. The method is compared only against the open source reimplementation of competitors, which clearly underperform compared to the results reported in the respective papers. The authors clearly report this in the paper, however I think it would be more fair to also report some qualitative comparison to other methods using the same prompts and results from the respective papers. This is what most published method have done to present their results (e.g., Dreamfusion, Magic3D, Fantasia3D). Moreover works like TextMesh and Magic3D on top of the qualitative results report user studies against the publicly available models from the Dreamfusion online gallery. The authors report a user preference study only against the (weaker) open source baseline used for their work. This is striking comparing the results for ProlifcDreamer reported by this work and the one reported in the original paper that seems to be significantly better,
2. The paper does not incorporate any quantitative evaluation of the quality of the generated 3D assets. Unfortunately this is a trend that many recent works in the field are following but earlier published works like DreamFusion do incorporate some quantitative measurement that might help to quantify advancement on less cherry picked samples.

b.**Generation might be too constrained by the retrieved assets**: The main contribution of the method is the incorporation of retrieved assets from Objaverse into the optimization objective to help create more plausible 3D models. While this clearly helps in the generation of object centric 3D models it’s unclear how the proposed models would generalize to scenes generation, which the baseline ProlificDreamer seems to be able to handle up to some extent (see Fig.1 (b) from the [ProlificDreamer paper](https://arxiv.org/pdf/2305.16213.pdf)). Moreover from the current examples in the work it’s unclear how much the proposed model is “generating” novel objects and how much it’s just “copying and retexturing” assets in Objaverse. It would have been great if the authors had included for each result also the closest Objaverse objects used during the optimization process similar to what was done in Fig. 11. Examples like “A fierce tiger” or “A fighter aircraft” in Fig. 7 might be already faithfully covered by objects in Objaverse.

c. **Relying on CLIP for aligning 3D assets**: In Sec. 4.4 the authors mention that they rely on CLIP to identify which one is the “front view” of the 3D assets by rendering several views and computing the cosine distance against prompts like “front/side/back view”. In my opinion this has two important limitations, first of all assumes that objects have a front/back/side view while whole categories of objects don’t have these properties (e.g., any symmetric object like a flower pot or a mug). Second it assumes that CLIP can be used successfully for this task, while to the best of my knowledge this has not been clearly shown before (and per my experience does not really work reliably). I would be curious to know what the experience of the authors is in using CLIP to identify camera poses. Also the fact that the models were aligned using CLIP helps to maximize the performance according to the evaluation protocol used in Fig. 9, since basically the seed shapes from objaverse were aligned to maximize the CLIP alignment computed there. If the model would just copy the objaverse example aligned with the proposed method the metric reported in Fig. 9 will be perfect but will not really measure anything useful related to the generative capabilities of the model.


## Minor

d. **Clarity**: In general I think the paper could be rewritten and made more clear. For example the English throughout the document can be improved, Fig. 9 does not have any kind of scale on the y axis, Fig. 14 caption is wrong.

e. **No discussion of limitations**: the paper does not discuss limitations and failure mode of the proposed method.

**Questions:**

Can you comment on the point I raised in weakness (a)-(b)-(c) and correct any misunderstanding I might have on the contributions of the paper?

Also see weakness (d) for small suggestions on how to improve the paper.

**Details Of Ethics Concerns:**

The paper contains a user study to evaluate the perceived quality of the generated assets but doesn't share many details on how the study was set up, how participants to the user study were selected and if they were compensated for the study or not.

---

### Official Review · Reviewer_3ZvE · 2023-11-01

**Soundness:** 3 good
**Presentation:** 3 good
**Contribution:** 3 good
**Rating:** 5
**Confidence:** 4

**Summary:**

In this paper, the authors tackle the challenge of inconsistencies in the generated 3D scenes caused by Score Distillation Sampling (SDS) based on 2D diffusion models. To address this, the authors propose a retrieval-augmented text-to-3D generation: They utilize a particle-based variational inference framework and enhance the conventional target distribution in SDS-based techniques with an empirical distribution of retrieved 3D assets. Additionally, they introduce an adapted 2D prior model to reduce bias towards certain camera viewpoints and delta distillation to regularize artifacts in generated 3D content. Experimental results demonstrate that their method improves geometry compared to the baseline.

**Strengths:**

1. The paper is clearly written and well-motivated. Given the large-scale 3d datasets nowadays, they show how to effectively use them and overcome the inconsistency problems caused by the SDS loss.

2. The qualitative results show improvements regarding the geometry compared to the baselines, while some more cases should be included.

**Weaknesses:**

I have a significant concern regarding the comprehensiveness of the experimental results, as they seem to lack crucial details:

a) The paper doesn't address scenarios where the text prompt is highly innovative, leading to a nearest neighbor that doesn't align well with the text prompt. How does an imperfect or conflicting retrieved object impact the final 3D generation?

b) The quantitative evaluation needs strengthening. Details such as the number of scenes used in the user study and the number of questions each participant answered are essential. What's more, there is no official implementation of ProlificDreamer currently, and the threestudio version is very unstable. The user study should cover more methods, especially those with official implementations.

c) The paper should also provide more qualitative results, particularly when the quantitative data might not be as reliable. Displaying more uncurated cases and separately showcasing the geometry (normal map) would offer a more comprehensive understanding of the model’s performance.

**Questions:**

See weakness.